# Regularization Neural Networks via Constrained Virtual Movement Field

## Abstract

We provide a novel thinking of regularization neural networks. We smooth the objective of neural networks w.r.t small adversarial perturbations of the inputs. Different from previous works, we assume the adversarial perturbations are caused by the movement field. When the magnitude of movement field approaches 0, we call it virtual movement field. By introducing the movement field, we cast the problem of finding adversarial perturbations into the problem of finding adversarial movement field. By adding proper geometrical constraints to the movement field, such smoothness can be approximated in closed-form by solving a min-max problem and its geometric meaning is clear. We define the approximated smoothness as the regularization term. We derive three regularization terms as running examples which measure the smoothness w.r.t shift, rotation and scale respectively by adding different constraints. We evaluate our methods on synthetic data, MNIST and CIFAR-10. Experimental results show that our proposed method can significantly improve the baseline neural networks. Compared with the state of the art regularization methods, proposed method achieves a tradeoff between accuracy and geometrical interpretability as well as computational cost.

## 1 Introduction

Deep neural networks have achieved great success in recent years Lecun et al. (2015). By improving the depth of computational graphs and the accounts of trainable parameters, neural networks can fit the training dataset better. However, overfitting becomes a serious problem in supervised training especially when the free parameters are numerous.

One of the most effective ways to against overfitting is adding regularization terms into the original supervised objective function. Many regularization methods have been proposed for training neural networks, such as dropout Srivastava et al. (2014) and its variants Wang & Manning (2013); Kingma et al. (2015). From a Bayesian perspective, dropout regularizes neural networks by introducing randomness into the parameters. Another regularization way to against overfitting is generating new data by transform or perturbation the existing data. The objective of the generated data or the smoothness w.r.t the small perturbations can be regarded as a regularization term. Bachman et al. (2014) assume the perturbations are fully random. They use the model's sensitivity to those random perturbations in their construction of the regularization function. However, Goodfellow et al. (2015); Szegedy et al. (2014) found the robustness of neural networks can't be improved sufficiently with random noise. Instead of using random perturbations, adversarial training (AT) Goodfellow et al. (2015) and virtual adversarial training (VAT) Miyato et al. (2016) find the so-called adversarial perturbations by optimizing some objectives under simple constraints, such as $L_\infty$ norm and $L_2$ norm. Specifically, AT selects the adversarial perturbation direction which maximizes the objective of neural networks. This lead to set the direction of perturbation the same as the gradients of objective w.r.t the inputs. Once the optimal perturbation is obtained, they apply it to the inputs and get the perturbated inputs. Then AT minimizes the objective of both the original inputs and the perturbated inputs. VAT follows similar spirits of AT. The key difference is that VAT obtains the optimal perturbation by maximizing Kullback-Leibler divergence (KLD) between the outputs of models. This makes VAT applicable for semi-supervised learning. VAT also designs an iterative algorithm to approximate the optimal perturbation. Experimental results demonstrate that AT and VAT against the adversarial perturbation as well as improve the generalization ability of neural networks. However, there are two drawbacks of AT and VAT.

- For a single batch of data, both AT and VAT need to run at least two forward-backward loops to complete the training process, which is time-consuming for big models.

- The obtained optimal perturbation lacks geometrical interpretability. It is hard to fully understand why those adversarial perturbations are most likely to fool neural networks.

We try to overcome the above drawbacks by introducing constraints into the space of perturbations. In this work, we assume *perturbations are caused by the movement field of the lattice structured data*, such as speech signals, images and videos. Movement field represents the motion vector of each pixel in the lattice. We call it virtual movement field when the magnitude of the movement field is sufficiently small. We smooth the objective of neural networks when the virtual movement field is applied to the inputs. Inspired by VAT, we first find the so-called adversarial movement field which maximizes above smoothness under a set of constraints. Then we minimize the corresponding smoothness under the adversarial virtual movement field. That is to obtain the smoothness, we need to solve a min-max problem with constraints. This can be done in closed-form if the set of constraints are carefully designed. Once this smoothness term is obtained, we minimize the objective of neural networks together with it. Because the movement field is virtual, it is unnecessary to get the perturbated inputs. Instead, the smoothness is expressed by the derivative of the objective w.r.t the movement field. Thus, the training process of each batch is completed in a single forward-backward loop which yields lower computational costs. Moreover, the obtained perturbations or smoothness term are much more interpretable because of the constraints. For example, we can see which direction of movement of an image is most likely to fool neural networks. We call our method as *virtual movement training* (VMT). We summarize the novelties of VMT as follows:

- The assumption of "small perturbations are caused by the virtual movement field" is a completely new idea in the literature of adversarial training or adversarial examples. By this assumption, we introduce data dependent constraints into the space of perturbations. And we cast the problem of finding perturbations into the problem of finding movement field.

- We develop a general framework to design regularization terms for neural networks trained with lattice structured data, i.e. solving a min-max problem associated with the movement field. closed-form terms are obtained by introducing proper geometrical constraints to the movement field.

In this work, we focus more on computational efficiency and geometrical interpretability of our method instead of against adversarial examples Szegedy et al. (2014). We derive three simple regularization terms as running examples based on introducing different constraints (shift, rotation and scale) into movement fields. These regularization terms measure how sensitive of neural networks under virtual (extremely small) shift, rotation and scale perturbations respectively.

We evaluate our method in a 1D synthetic data and two benchmark image classification datasets: MNIST and CIFAR-10. Experimental results demonstrate that our method remarkably improves the baseline neural networks. Compared with AT and VAT, VMT achieves a tradeoff between accuracy and geometrical interpretability as well as computational cost.

## 2 METHODS

We first formally define the movement field and the virtual movement field. Then we formulate our method. Finally, we provide three running examples.

### 2.1 VIRTUAL MOVEMENT FIELD

For data $I \in \mathcal{R}^{d_1 \times d_2 \cdots \times d_n}$, i.e. $n$ dimension lattice structure and the length of $i$th dimension is $d_i$, we define the movement field $V$ of as an $n + 1$ dimension tensor, that is $V \in \mathcal{R}^{d_1 \times d_2 \cdots \times d_n \times n}$. Denote $p \in \mathcal{Z}^n$ as the position vector of $I$ and $I_p$ is the value in that position. Then $V_p \in \mathcal{R}^n$ is the movement of location $p$, i.e. its new position would be $p + V_p$. Note that for 2 dimension lattice data such as images, their movement field is somewhat similar to the concept of optical flow. However, throughout this paper, we still use the word of "movement field" because it is generalized to any dimension of lattice data. If we assume data $I$ is sampled from an underlying continues space or

the first order derivatives of $I$ exists, we can approximate the value of the new position with the first order Taylor series of the value of the original position (when the movement is small). Formally

$$I_{p+V_p} = I_p + \left(\frac{\partial I_p}{\partial p}\right)^T V_p \tag{1}$$

For $V_p$, there are two factors: the length and the direction. In some cases, it is necessary to decompose those two factors. So we normalize it as follows:

$$\widetilde{V}_p = \frac{V_p}{Z}, \quad Z = \sqrt{\mathbf{E}[V_p^T V_p]} \tag{2}$$

Then the averege square length of $\widetilde{V}$ is equal to one. Denote $\varepsilon \widetilde{V}$ as the actual movement field. We call $\varepsilon$ the degree of the movement field. When $\varepsilon$ approaches $0$, we call $\widetilde{V}$ *the virtual movement field*. Based on (1) and (2), if $\widetilde{V}$ is given, we have:

$$\frac{\partial I_p}{\partial \varepsilon} = \left(\frac{\partial I_p}{\partial p}\right)^T \widetilde{V}_p \tag{3}$$

## 2.2 PROBLEM FORMULATION

Given a dataset $\mathcal{D} = \{(I^n, y^n)|n = 1, 2, \ldots, N\}$, where $I^n$ and $y^n$ are $i$th pair of input and label in $\mathcal{D}$. Denote $f_\theta$ as a function which is parameterized by $\theta$. $f_\theta$ maps the input space into the output space. For each pair of $\{I^n, y^n\}$, we minimize the predefined loss function between the predicted output and the label w.r.t $\theta$.

$$\arg\min_\theta \mathcal{L}(I^n, y^n) \tag{4}$$

We expect that $\mathcal{L}$ is stable for some particular kind of movements of $I$, e.g. rotation for images. We can apply a small movement $\varepsilon \widetilde{V}$ to $I^n$ and we get the new input $I^n(\varepsilon \widetilde{V})$. Intuitively, since $\widetilde{V}$ is normalized we can measure the smoothness of $\mathcal{L}$ under the movement field $\widetilde{V}$ as follows:

$$\left|\frac{\mathcal{L}(I^n(\varepsilon\widetilde{V}), y^n) - \mathcal{L}(I^n, y^n)}{\varepsilon}\right| \tag{5}$$

That is the proportion between the change of objective and the degree of inputs movement. When training neural networks, in order to obtain the above smoothness, we need to run the forward computational graph two times: the one for $\mathcal{L}(I^n, y^n)$, the other for $\mathcal{L}(I^n(\varepsilon\widetilde{V}), y^n)$. However, in this work, we are interested in the extreme situation of (5): what if we apply a virtual movement field to $I_p$? Or how sensitive the objective w.r.t $\varepsilon$ when $\varepsilon \to 0$. Note that $\mathcal{L}$ can be reparameterized as a function of $\varepsilon$ by fixing other variables. Thus when $\varepsilon \to 0$, (5) is equivalent to

$$\lim_{\varepsilon \to 0}\left|\frac{\mathcal{L}(\varepsilon) - \mathcal{L}(0)}{\varepsilon}\right| = \left|\frac{\partial \mathcal{L}}{\partial \varepsilon}\Big|_{I^n, y^n, \theta, \widetilde{V}}\right| \tag{6}$$

By the chain rule for differentiation

$$\frac{\partial \mathcal{L}}{\partial \varepsilon} = \sum_p \frac{\partial \mathcal{L}}{\partial I_p}\frac{\partial I_p}{\partial \varepsilon} \tag{7}$$

Substitute (3) into it, we have

$$\frac{\partial \mathcal{L}}{\partial \varepsilon} = \sum_p \frac{\partial \mathcal{L}}{\partial I_p}\left(\frac{\partial I_p}{\partial p}\right)^T \widetilde{V}_p \tag{8}$$

The first term on the right side of the above equation is the gradient of objective w.r.t the input which is easily obtained by back propagation. The second term is the gradient of input w.r.t its coordinates which is also called the directional derivative of signals. For discrete signal, it is approximated by finite difference operator in practice. Thus, once $\widetilde{V}$ is given, we can obtain the smoothness of objective w.r.t $\varepsilon$ when the corresponding virtual movement field applied to the input in a single forward-backward loop.

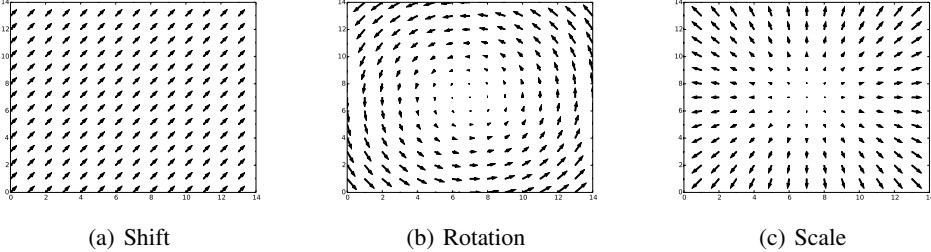

|                     (a) Shift                     |                   (b) Rotation                   |                   (c) Scale                    |

Figure 1: Three kinds of movement fields: shift, rotation and scale are shown in (a), (b) and (c) respectively. These fields are generated in a $15 \times 15$ image.

Table 1: Summarization of movement fields and their corresponding regularization terms. The meaning of symbols can be found in section 2.2 and 2.3.

$$\widetilde{V}_p^{shift} = \begin{pmatrix} \cos \phi \\ \sin \phi \end{pmatrix} \qquad \mathcal{R}_{shift} \overset{\text{def}}{=} \sqrt{\left( \sum_p \frac{\partial \mathcal{L}}{\partial I_p} \frac{\partial I_p}{\partial p_1} \right)^2 + \left( \sum_p \frac{\partial \mathcal{L}}{\partial I_p} \frac{\partial I_p}{\partial p_2} \right)^2}$$

$$\widetilde{V}_p^{rotation} = \frac{1}{Z} \begin{pmatrix} -p_2 - c_1 \\ p_1 - c_2 \end{pmatrix} \qquad \mathcal{R}_{rotation} \overset{\text{def}}{=} \left| \sum_p \frac{\partial \mathcal{L}}{\partial I_p} \left( \frac{\partial I_p}{\partial p}^T \right) \widetilde{V}_p^{rotation} \right|$$

$$\widetilde{V}_p^{scale} = \frac{1}{Z} \begin{pmatrix} (p_1 - c_1) \cos \phi \\ (p_2 - c_2) \sin \phi \end{pmatrix} \qquad \mathcal{R}_{scale} \overset{\text{def}}{=} \frac{1}{Z} \sqrt{\left( \sum_p \frac{\partial \mathcal{L}}{\partial I_p} \frac{\partial I_p}{\partial p_1} (p_1 - c_1) \right)^2 + \left( \sum_p \frac{\partial \mathcal{L}}{\partial I_p} \frac{\partial I_p}{\partial p_2} (p_2 - c_2) \right)^2}$$

However, the degree of freedom of $\widetilde{V}$ is extremely high for real-world data. Search for such high dimension is inefficient. Thus it is necessary to introduce constraints into $\widetilde{V}$. Those constraints should embody the priors of data and physical mechanisms of how it is generated. For example, the movement field of natural images should enjoy the properties of local smoothness and isotropy. We denote the set of constraints as $\mathcal{C}(\widetilde{V})$. Note that the normalization constraint in (2) is included in $\mathcal{C}(\widetilde{V})$. We can expect that the degree of freedom of $\widetilde{V}$ is sufficiently reduced under those constraints. If there are still freedom of $\widetilde{V}$, we can randomly draw samples over those freedom. However, inspired by the adversarial training, we first find the adversarial movement field $\widetilde{V}^*$ which maximizes the smoothness of neural networks then we minimize the obtained smoothness term plus the original objective w.r.t $\theta$ under $\widetilde{V}^*$. Similar to the generative adversarial networks (GAN) Goodfellow et al. (2014), above problem can be formulated as a min-max game under constraints:

$$\min_{\theta} \max_{\widetilde{V}} \quad \mathcal{L}(I, y) + \lambda \left| \frac{\partial \mathcal{L}}{\partial \varepsilon} \Big|_{I, \widetilde{V}, y, \theta} \right|, \quad s.t. \quad \mathcal{C}(\widetilde{V}) \tag{9}$$

Once $\widetilde{V}^*$ is obtained by solving the above max problem, the second term in (9) is determined. We call it as the corresponding regularization term of $\widetilde{V}^*$. Then (9) is reduced to

$$\min_{\theta} \quad \mathcal{L}(I, y) + \lambda \mathcal{R}(\widetilde{V}^*) \tag{10}$$

Generally, solving the max problem in (9) is not an easy task. However, we will show that $\widetilde{V}^*$ can be obtained in closed-form if the constraint set is carefully designed. And in this paper, we just focus on this simple case because we hope to train each batch of data in a single forward-backward loop.

### 2.3 DESIGN THE MOVEMENT FIELD

Now we provide three sets of constraints for $\widetilde{V}$ which make the corresponding $\widetilde{V}^*$ solved in closed-form. All these movement fields are designed for 2D lattice data since image is one of the most important types of data in real world.

The first one is called Shift field. That is all pixels in 2D lattice are shifted by the same vector. Because $\widetilde{V}$ is normalized, the only freedom is the direction of the vector in 2D space. Formally

$$\widetilde{V}_p^{shift} = (\cos \phi, \sin \phi)^T, \quad \forall p \tag{11}$$

Combination (11) and (8), the max problem in (9) is equivalent to

$$\max_{\phi} \left| \left( \sum_p \frac{\partial \mathcal{L}}{\partial I_p} \frac{\partial I_p}{\partial p_1} \right) \cos \phi + \left( \sum_p \frac{\partial \mathcal{L}}{\partial I_p} \frac{\partial I_p}{\partial p_2} \right) \sin \phi \right| \tag{12}$$

where $p_1$ and $p_2$ are the first coordinate and the second coordinate of an image respectively. Then the optimal value of $\phi$ is obtained easily. And the corresponding maximum value of $|\partial \mathcal{L}/\partial \varepsilon|$ in this case is

$$\mathcal{R}_{shift} \overset{\text{def}}{=} \sqrt{\left( \sum_p \frac{\partial \mathcal{L}}{\partial I_p} \frac{\partial I_p}{\partial p_1} \right)^2 + \left( \sum_p \frac{\partial \mathcal{L}}{\partial I_p} \frac{\partial I_p}{\partial p_2} \right)^2} \tag{13}$$

See Appendix C for derivation. The second movement field is rotation field. In this work, we simply assume the center of rotation is the center of the image, i.e. $(c_1, c_2)$.

$$\widetilde{V}_p^{rotation} = \frac{1}{Z}(-p_2 - c_1, p_1 - c_2)^T \tag{14}$$

where $Z$ is the normalization constant described in (2). Thus the degree of freedom is 0 (The direction of rotation doesn't matter because we care about the absolute value of $\partial \mathcal{L}/\partial \varepsilon$). Then $\mathcal{R}_{rotation}$ is obtained straightforwardly.

The third movement field is scale field which scales an image by different factors along two coordinates. Thus the degree of freedom is 1. We parameterize it by

$$\widetilde{V}_p^{scale} = \frac{1}{Z}((p_1 - c_1) \cos \phi, (p_2 - c_2) \sin \phi)^T \tag{15}$$

Then $\mathcal{R}_{scale}$ is obtained in a similar way as $\mathcal{R}_{shift}$. We summary these three movement fields and their corresponding derived regularization terms in Tab 1. Although other kinds of movement fields are possible to be designed, we just use them to evaluate our method because they are simple, easy to implementation and geometrically meaningful.

## 2.4 PRACTICAL CONSIDERATIONS

As mentioned in section 2.2, for lattice data, the directional gradients are approximated by finite difference operator. In this work, we choose the simplest one:

$$\frac{\partial I_x}{\partial x} \approx \frac{I_{x+1} - I_{x-1}}{2} \tag{16}$$

where $x$ is an arbitrary coordinate in $p$. If the local smooth property is not well satisfied, above approximation is not accurate. This suggests that our method is more suitable for smooth data.

Another problem is that we find the magnitudes of $\partial \mathcal{L}/\partial I$ change rapidly with the network configurations. This makes the values of the corresponding regularization terms change rapidly with the network configurations. To keep the values of $\mathcal{R}$ in a stable range, we normalize $\partial \mathcal{L}/\partial I$ into a unit tensor.

## 3 DISCUSSIONS AND RELATED WORKS

Our work was mainly motivated by the adversarial training Goodfellow et al. (2015) and was related to the virtual adversarial training Miyato et al. (2016). Adversarial training can be reformulated as follows:

$$\min_{\theta} \max_{\delta I} \mathcal{L}(I, y) + \lambda \mathcal{L}(I + \varepsilon \delta I, y), \quad s.t. \quad ||\delta I||_{\infty} < 1 \tag{17}$$

and $\delta I$ is approximated by $\mathbf{sign}(\partial \mathcal{L}/\partial I)$ in their paper because the only constraint of $\delta I$ is $L_{\infty}$ norm. However, in our work, we assume $\delta I$ is caused by the movement field $\widetilde{V}$.

$$I(\varepsilon \widetilde{V}) \to I + \varepsilon \delta I \tag{18}$$

That is the perturbation $\delta I$ is constrained by both the movement field $\widetilde{V}$ and the directional gradients of $I$, instead of the simple norm constraint. Another key difference between AT and our method is

$\varepsilon \to 0$ in our work, thus it is unnecessary to generate $I + \varepsilon\delta I$ and run additional forward-backward loop. By setting $\varepsilon$ sufficiently small, (17) is equivalent to

$$\min_{\theta} \max_{\delta I} \quad (1 + \lambda)\mathcal{L}(I, y) + \lambda\varepsilon\frac{\partial\mathcal{L}}{\partial\varepsilon}\Big|_{I,\delta I,y,\theta} \tag{19}$$

This formulation is similar to ours which suggests our method is an extreme case of AT if we ignore the difference between constraints of perturbations.

Denote $f_{\theta}$ as the forward function of neural networks parameterized by $\theta$. Then virtual adversarial training can be reformulated as follows:

$$\min_{\theta} \max_{\delta I} \mathcal{L}(I, y) + KLD[f_{\theta}(I)||f_{\theta}(I + \varepsilon\delta I)], \quad s.t. \quad ||\delta I||_2 < 1 \tag{20}$$

The core difference between VAT and AT is that VAT minimizes KLD of the outputs of neural networks under adversarial perturbations. This property makes VAT applicable for semi-supervised learning. However, the KLD term makes it difficult to find the optimal perturbation. Thus Miyato et al. (2016) developed an iterative algorithm for approximation.

The idea of smooth the objective w.r.t small perturbations also has appeared in earlier works, such as tangent propagation Simard et al. (1998) and influence function Koh & Liang (2017). The derivative of the objective w.r.t perturbations is obtained by chain rule. However, the transformations are predefined in Simard et al. (1998) while the transformations in VMT are obtained by solving a constrained min-max problem though the freedom of those transformations is low currently. For influence function, there are no constraints in the space of perturbations. And the smoothness w.r.t small perturbations is mainly used to analyze the behaviors of a trained model instead of regularizing it during training.

In summary, we focus more on geometrical interpretability and computational efficiency of our method. Thus our method is not required to be better than VT and VAT. However, we still compare them in next section.

## 4 EXPERIMENTAL RESULTS

We evaluate our proposed method for supervised classifications in three datasets: 1D synthetic dataset, MNIST and CIFAR-10. We compare it with the baseline, adversarial training with $L_2$ constraint and virtual adversarial training. For each dataset, the shared hyper-parameters keep same for all methods. While separate hyper-parameters are tuned by cross-validation using grid-search or copied from literature if they are provided. $\varepsilon$ in (17) and (20) is searched over $[0.01, 10]$ and $\lambda$ in (10) is searched over $[0.005, 5]$. All neural networks are implemented in Tensorflow Abadi et al. (2015). We call our method VMT-shift, VMT-rotation and VMT-scale when we use the corresponding regularization terms.

### 4.1 THE BINARY CLASSIFICATION OF 1D SYNTHETIC DATASET

We create a 1D synthetic dataset with two classes using the following random process:

$$x = \sin(\omega t + \phi) + \eta \tag{21}$$
$$\phi \sim \mathcal{U}(0, \frac{\pi}{2})$$
$$\eta \sim \mathcal{N}(0, 0.1^2)$$

where $t \in \mathcal{R}^{100}$ is uniformly sampled from $[-2\pi, 2\pi]$. Thus $x \in \mathcal{R}^{100}$ is a 1D lattice signal. Based on (21), we generate 5000 positive samples by setting $\omega = 0.99$ and 5000 negative samples by setting $\omega = 1.01$. We randomly select 1000 samples as training set and the rest as test set.

We train neural networks with two hidden layers each of which has 128 units and is followed by batch normalization Ioffe & Szegedy (2015) and ReLU activation Glorot et al. (2011). We set batchsize to 20 and run 100 epochs using ADAM optimizer Kingma & Ba (2015). We run each method 5 times and report the average test errors.

We summarize the results in Tab 2. VMT-shift is significantly better than the baseline and is slightly better than AT. We show values of $\mathcal{R}_{shift}$ on test set over epochs in Fig 2(b). The values of $\mathcal{R}_{shift}$

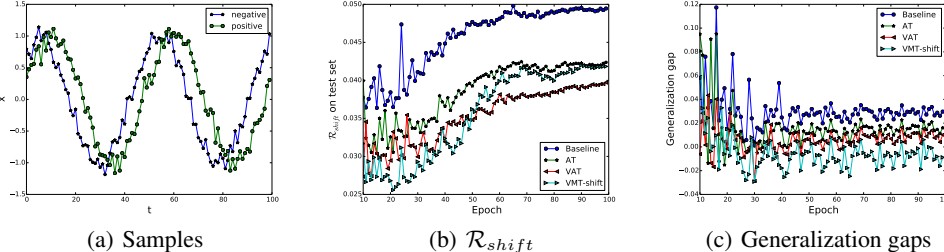

(a) Samples          (b) $\mathcal{R}_{shift}$          (c) Generalization gaps

Figure 2: (a) shows an example pair of data with different labels. They are hard to distinguish by human-eye. (b) shows the values of $\mathcal{R}_{shift}$ on test set over epochs. (c) shows the generalization gaps for different methods. The values in the first 10 epochs are omitted for better visualization.

are consistent with the test errors. That is the smaller the values of $\mathcal{R}_{shift}$ at the end of training, the smaller the test errors. We also empirically compare the generalization abilities of those methods. The generalization ability is measured by the so-called generalization gap, i.e. the difference between the empirical risks on test set and training set.

$$\mathbf{E}_{(I,y)\in\mathcal{D}_{test}}\mathcal{L}(I,y) - \mathbf{E}_{(I,y)\in\mathcal{D}_{train}}\mathcal{L}(I,y) \tag{22}$$

It is suggested by Fig 2(c) that $\mathcal{R}_{shift}$ is an appropriate regularization term which can improve the generalization ability of baseline on our hand-created data.

We find that VMT-scale is inferior to baseline on our hand-created data. This is not surprising because each sample is labeled by its angular frequency, i.e. $\omega$. When we scale a sample, its angular frequency will be changed. Thus the assumption of the smoothness under small scale perturbations is not satisfied in this case. In fact, the priors of data can be encoded in the constraints of movement field. Thus, for VMT, it is possible to targeted design regularization terms based on the properties of data.

### 4.2 THE CLASSIFICATION OF MNIST DATASET

We tested the performance of our regularization method on the MNIST dataset, which consists of handwritten digits with size $28 \times 28$ and their corresponding labels from 0 to 9. We train our models using the whole 60000 training samples. For network structures, we follow the setting used in Miyato et al. (2016). Specifically, we train NNs with 4 hidden dense layers with nodes $(1200, 600, 300, 150)$ respectively. Each hidden layer is followed by batch normalization and ReLU activation.

We apply all of the three regularization terms in Tab 1 and compare them with the baseline, dropout, AT and VAT. We run each method 5 times and report the average test errors. The results are summarized in Tab 2 which show our methods are inferior to AT and VAT but are significantly better than baseline.

The training time on Synthetic dataset and MNIST is summarized in Appendix B.

### 4.3 THE CLASSIFICATION OF CIFAR-10 DATASET

We also conducted studies on the CIFAR-10 dataset (without data argument) which consists of 50000 training images and 10000 testing images in 10 classes, each image with size $32 \times 32 \times 3$. Our focus is on the behaviors of different regularization methods, but not on pushing the state-of-the-art results, so we use a relatively small neural network for evaluation and comparison of those regularization methods. Specifically, we configure the neural networks as the "conv-small" used in Salimans et al. (2016) which contains 9 convolutional layers. See appendix A for detailed architecture. All neural networks are trained by SGD with momentum with 80 epochs. We evaluate the test errors and the average training time of each epoch. We define $\mathcal{R}_{all}$ as the linear combination of the regularization terms in Tab 1. Formally

$$\mathcal{R}_{all} = \frac{w_1\mathcal{R}_{shift} + w_2\mathcal{R}_{rotation} + w_3\mathcal{R}_{scale}}{w_1 + w_2 + w_3} \tag{23}$$

Table 2: Test errors (%) on MNIST and synthetic dataset. Test errors in the bottom panel are the results of our methods. Test errors with "*" are the ones reported in the literature.

| Methods | Synthetic | MNIST |
|---|---|---|
| Baseline | 1.24 | 1.12 |
| Dropout Srivastava et al. (2014) | 0.97 | 0.95* |
| AT(with $L_\infty$ constraint) Goodfellow et al. (2015) | 1.00 | 0.72 |
| AT(with $L_2$ constraint) Goodfellow et al. (2015) | 0.94 | 0.70 |
| VAT Miyato et al. (2016) | 0.76 | 0.64 |
| VMT-shift | 0.89 | 0.92 |
| VMT-rotation | - | 0.95 |
| VMT-scaling | - | 0.92 |

Table 3: Test errors (%) and running time (s) on CIFAR-10. Running time in this table means the average training time of each epoch.

| Methods | Test errors | Time |
|---|---|---|
| Baseline | 10.79 | 27.23 |
| AT(with $L_2$ constraint)Goodfellow et al. (2015) | 10.42 | 55.48 |
| VAT Miyato et al. (2016) | 9.62 | 59.95 |
| VMT-shift | 9.68 | 37.55 |
| VMT-rotation | 9.75 | 37.57 |
| VMT-scale | 9.74 | 37.59 |
| VMT-all | 9.31 | 37.68 |

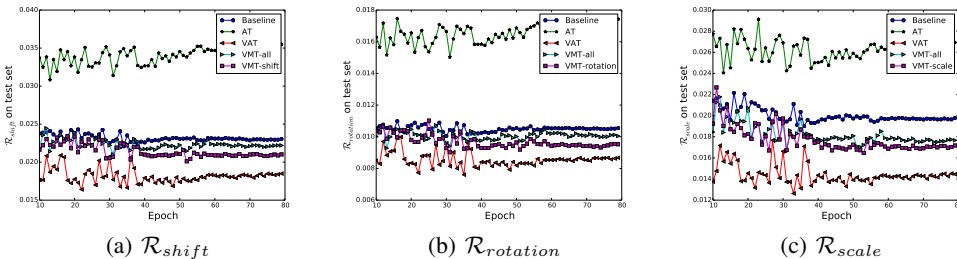

(a) $\mathcal{R}_{shift}$     (b) $\mathcal{R}_{rotation}$     (c) $\mathcal{R}_{scale}$

Figure 3: (a), (b) and (c) show the values of $\mathcal{R}_{shift}$, $\mathcal{R}_{rotation}$ and $\mathcal{R}_{scale}$ respectively for different methods on test set over epochs.

where $w_i \sim \mathcal{U}(0, 1)$. We randomly sample $w_i$ for each batch. VMT-all means we use $\mathcal{R}_{all}$ as the regularization term. Results are summarized in Tab 3. All our regularization terms are significantly better than the baseline and AT. And they are competitive compared with VAT. When we linearly combine $\mathcal{R}_{shift}$, $\mathcal{R}_{rotation}$ and $\mathcal{R}_{scale}$, the performance is further improved by a relatively large margin. Such combination is cheap in practice (See running time in Tab 3). We believe that the performance can be improved to a higher level if we design and combine more regularization terms. And our method is faster than AT and VAT. We show the values of the regularization terms in Fig 3. All these values of our methods are smaller than the values of baseline.

## 5 CONCLUSIONS

In this paper, we have provided a novel thinking of regularization neural networks. We smooth the objective function of neural networks when the virtual moment field is applied to lattice data. By carefully introducing constraints into the movement field, we have derived the smoothness in closed-form by solving a min-max problem. We have provided three regularization terms which measure the smoothness w.r.t the transformations of shift, rotation and scale respectively. Experimental results demonstrate that our method remarkably improves the baseline neural networks on 1D synthetic data, MNIST and CIFAR-10. Compared with AT and VAT, VMT achieves a tradeoff between accuracy and geometrical interpretability as well as computational cost. Unlike AT and VAT, the training process of each batch is completed in a single forward-backward loop. Moreover, by control the movement field, we can understand the geometric meaning of perturbations and what kind of smoothness the regularization term is measured.

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

# A    DETAILED ARCHITECTURE OF "CONV-SMALL"

Table 4: CNN model used in our experiments on CIFAR-10 whose architecture are same as the architecture of "conv-samll" in Salimans et al. (2016). The slopes of all LeakyReLU (LReLU) functions Maas et al. (2013) are set to 0.1.

| |
|---|
| $3 \times 3$ conv. $64 \rightarrow$ batch normalization $\rightarrow$ LReLU |
| $3 \times 3$ conv. $64 \rightarrow$ batch normalization $\rightarrow$ LReLU |
| $3 \times 3$ conv. $64 \rightarrow$ batch normalization $\rightarrow$ LReLU |
| $2 \times 2$ max pooling |
| dropout, $p = 0.5$ |
| $3 \times 3$ conv. $128 \rightarrow$ batch normalization $\rightarrow$ LReLU |
| $3 \times 3$ conv. $128 \rightarrow$ batch normalization $\rightarrow$ LReLU |
| $3 \times 3$ conv. $128 \rightarrow$ batch normalization $\rightarrow$ LReLU |
| $2 \times 2$ max pooling |
| dropout, $p = 0.5$ |
| $3 \times 3$ conv. $128 \rightarrow$ batch normalization $\rightarrow$ LReLU |
| $1 \times 1$ conv. $128 \rightarrow$ batch normalization $\rightarrow$ LReLU |
| $1 \times 1$ conv. $128 \rightarrow$ batch normalization $\rightarrow$ LReLU |
| global average pooling |
| dense. $10 \rightarrow$ softmax |

# B    TRAINING TIME ON SYNTHETIC DATA AND MNIST

Table 5: Average training time (s) of each epoch on Synthetic data and MNIST. Since the training time of VMT-shift, VMT-rotation and VMT-shift are almost the same, we use their averaged training time.

| Dataset | baseline | AT-$L_2$ | VAT | VMT |
|---|---|---|---|---|
| Synthetic | 0.183 | 0.288 | 0.329 | 0.279 |
| MNIST | 3.635 | 6.170 | 7.122 | 5.950 |

# C    DERIVATION OF $\mathcal{R}_{shift}$

By setting

$$a = \sum_p \frac{\partial \mathcal{L}}{\partial I_p} \frac{\partial I_p}{\partial p_1}, \quad b = \sum_p \frac{\partial \mathcal{L}}{\partial I_p} \frac{\partial I_p}{\partial p_2}$$

(12) is equivalent to

$$\left| a \cos \phi + b \sin \phi \right| = \left| \sqrt{a^2 + b^2} \cos(\phi - \alpha) \right|$$

where $\alpha = \tan^{-1} \frac{b}{a}$. Thus the maximum value of (12) is $\sqrt{a^2 + b^2}$ when $\phi = \alpha$.

