# OpenReview forum: "Regularization Neural Networks via Constrained Virtual  Movement Field"
_ICLR.cc/2018/Conference — Invite to Workshop Track_

### Official Review · AnonReviewer3 · 2017-11-28

**Rating:** 5
**Confidence:** 4

**Review:**

This paper proposes to regularize neural networks by the invariance to certain types of transforms. This is framed into a minimax problem, which yields a closed form regularization when constrained to simple types of transforms.

The basic idea of using derivative to measure sensitivity has been widely known, and is related to tangent propagation and influence function. Please comment on the connection and difference. What is the substantial novelty of this current approach?

The empirical results are not particularly impressive. The performance is not as good as  (and seems significantly worse than) AT and VAT on MNIST. Could you provide an explanation? On CIFAR10, VMT-all is only comparable with VAT. Although VMT is faster than VAT, it seems not a significant advantage since is not faster in a magnitude.

The writing need to be significantly improved. Currently there are lot of typos and grammar errors, e.g., \citep vs. \citet; randon, abouve, batchszie;  \mathcal{Z}^n is undefined when it first appears.

In VMT-all, how do you decide the relative importance of the three different regularizations?

Is Figure 3 the regularization on the training or testing set? Could you explain why it reflects generalization ability?

---

> ### Author Response · Authors · 2017-12-17
> **Authors' response to AnonReviewer3**
>
> Thanks for your comments.
> (1) For tangent propagation, the transformations are predefined while the transformations in VMT are defined by the movement field and are further obtained by solving a constrained min-max problem though the freedom of those transformations is low currently.
>
> For influence function, there are no constraints in the space of perturbations. And the smoothness w.r.t small perturbations is mainly used to analyze the behaviors of a trained model instead of regularizing it during training.
>
> The substantial novelties of our work are:
> * The assumption of “small perturbations are caused by the virtual movement filed” is a completely new idea in the literature of adversarial training or adversarial examples. By this assumption, we introduce data dependent constraints into the space of perturbations. And we cast the problem of finding perturbations into the problem of finding movement field.
>
> * We develop a general framework to design regularization terms for neural networks trained with lattice structured data, i.e. solving a min-max problem associated with the movement field. Close-form terms are obtained by introducing proper geometrical constraints to the movement field.
>
> We have added above content on the introduction section and the related work section in our updated paper.
>
> (2) Although our work is inspired by AT and VAT, it is not an incremental work of AT or VAT. It is unfair to say VMT must be better than AT or VAT. In fact, we focus more on geometrical interpretability and computational efficiency. Our method achieves a tradeoff between accuracy and those two factors compared with AT and VAT. As a regularization method, VMT achieves similar or better performance on synthetic dataset and MNIST compared with dropout, a widely used regularization technique for neural networks. This supports the effectiveness of VMT.
>
> As mentioned in the paper, VMT finishes the training process in a single forward-backward loop while AT and VAT need at least two forward-backward loops (two in practice). Thus it is impossible to be faster in a magnitude. However, we still think such reducing of computational cost is valuable when we train big neural networks. In fact, we think running time is a minor contribution to our work
>
> My personal view of the "bad" performance on MNIST is that: In VMT, we use finite difference to approximate the direction gradient of the inputs. This requires the local smooth property of the inputs. But for MNIST, we think the local smooth property is not well satisfied (it looks like binary values). AT and VAT do not rely on the local smooth property. Thus VMT is inferior to AT and VAT on MNIST.
>
> We believe our method can be further improved if we design more regularization terms by changing the constraints and combine those terms. Such combination is cheap in practice. See the performance and training time of VMT-all in Tab 3. Also, it is possible to targeted design regularization terms based on the properties of data.
>
> (3) We define Z in eq.(2) in our updated paper.
>
> (4) We randomly combine three terms in each batch. So, on average, the relative importance is equal. See eq.(23).
>
> (5) The values of Fig(3) are on test set. The word "generalization ability" is used unserious. What we want to say is: the values of all regularization methods are lower than baseline and the performance of all regularization methods is better than baseline. We measure "generalization ability" by the difference between the empirical risks on test set and training set in our updated paper. See section 4.1.

---

### Official Review · AnonReviewer1 · 2017-11-28
**The paper is well formalized and the idea is interesting. The regularization approach is novel compared to the methods of the literature. However, the experimental validation of the proposed approach is not consistent and the positioning of the proposed approach is not so clear. So, I  suggest to reject the paper in his actual form.**

**Rating:** 5
**Confidence:** 4

**Review:**

This paper tackles the overfitting problem when training neural networks based on regularization technique. More precisely, the authors propose new regularization terms that are related to the underlying virtual geometrical transformations (shift, rotation and scale) of the input data (signal, image and video). By formalizing the geometrical transformation process of a given image, the authors deduce constraints on the objective function which depend on the magnitude of the applied transformation. The proposed method is compared to three methods: one baseline and two methods of the literature (AT and VAT). The comparison is done on three datasets (synthetic data, MNIST and CIFAR10) in terms of test errors (for classification problems) and running time.

The paper is well formalized and the idea is interesting. The regularization approach is novel compared to the methods of the literature.

Main concerns:
1)	The experimental validation of the proposed approach is not consistent:
The description of the baseline method is not detailed in the paper.
A priori, the baseline should naturally be the method without your regularization terms.
But, this seems to be contrary with what you displayed in Figure 3.
Indeed, in Figure 3, there is three different graphs for the baseline method (i.e., one for each regularization term). It seems that the baseline method depends on the different kinds of regularization term, why? Same question for AT and VAT methods.
In practice, what is the magnitude of the perturbations?
Please, explain the axis of all the figures.
Please, explain how do you mix your different regularization terms in your method that you call VMT-all?
All the following points are related to the experiment for which you presented the results in Table 2:
Please, provide the results of all your methods on the synthetic dataset (only VMT-shift is provided). What is VMF? Do you mean VMT?
For the evaluations, it would be more rigorous to re-implement also the state-of-the-art methods for which you only give the results that they report in their paper. Especially, because you re-implemented AT with L-2 constraint, so, it seems straightforward to re-implement also AT with L-infinite constraint. Same remark for the dropout regularization technique, which is easy to re-implement on the dense layers of your neural networks, within the Tensorflow framework.
As you mentioned, your main contribution is related to running time, thus, you should give the running time in all experiments.

2)	The method seems to be a tradeoff between accuracy and running time:
The VAT method performs better than all your methods in all the datasets.
The baseline method is faster than all the methods (Table 3).
This being said, the proposed method should be clearly presented in the paper as a tradeoff between accuracy and running time.
3)	The positioning of the proposed approach is not so clear:
As mentioned above, your method is a tradeoff between accuracy and running time. But you also mentioned (top of page 2) that the contribution of your paper is also related to the interpretability in terms of ‘’Human perception’’. Indeed, you clearly mentioned that the methods of the literature lacks interpretability. You also mentioned that your method is more ‘’geometrically’’ interpretable than methods of the literature. The link between interpretability in terms of “human perception” and “geometry” is not obvious. Anyway, the interpretability point is not sufficiently demonstrated, or at least, discussed in the paper.

4)	Many typos in the paper :
Section 1: “farward-backward”
Section 2.1: “we define the movement field V of as a n+1…”
Section 2.2: “lable” - “the another” - “of how it are generated” – Sentence “Since V is normalized.” seems incomplete… - \mathcal{L} not defined - Please, precise the simplifications like \mathcal{L}_{\theta} to \mathcal{L}
Section 3: “DISCUSSTION”
Section 4.1: “negtive”
Figure 2: “negetive”
Table 2: “VMF”
Section 4.2: “Tab 2.3” does not exist
Section 4.3: “consists 9 convolutional” – “nerual networks”…
Please, always use the \eqref latex command to refer to equations.
There is many others typos in the paper, so, please proofread the paper…

---

> ### Author Response · Authors · 2017-12-17
> **Authors' response to AnonReviewer1**
>
> Thanks for your comments.
> 1) * For Fig.(3), x-axis means the number of training epoch. y-axis means the values of the regularization term on test set. The baseline is trained without any regularization term. But we can still evaluate the value of the corresponding regularization term of baseline on test set. Same for AT and VAT. Fig.(3a), Fig.(3b) and Fig.(3c) show the values of R_shift, R_rotation and R_scale respectively, no matter what regularization terms the model is trained with. So the baseline on each dataset is same. We describe the baseline on CIFAR-10 in Appendix A.
>
> * The magnitude of the perturbations changes over dataset. For AT and VAT, \varepsilon ranges from 0.01 to 10. For VMT, \lambda ranges from 0.005 to 5. We do grid-search over their range.
>
> * We explain how to mix regularization terms in eq.(23) in our updated paper.
>
> * VMT-rotation can't be applied to the synthetic dataset because there is no rotation operator for 1D signal. VMT-scale is not suitable for this dataset. We explain the reason at the end of section 4.1.
>
> * VMF means VMT. This is our writing mistake.
>
> * We don't re-implement AT-L_2 because the performance of AL-L_inf is slightly worse than AT-L_2 in previous literature. Now, we re-implement AT-L_inf on Synthetic dataset and MNIST. We also re-implement dropout on Synthetic dataset. For dropout on MNIST, we still use the result from literature. Because finding the optimal dropout rates for a 4-layer network requires lots of time and our preliminary results are inferior to the result from literature. So we think this result can approximate the best performance for dropout on MNIST.
>
> * We give the training time on Synthetic dataset and MNIST in appendix B. In fact, we think running time is a minor contribution to our work. See following comments.
>
> 2) Yes, currently, VMT is a tradeoff between accuracy and running time as well as geometrical interpretability. And it has been clearly presented in our updated paper.
>
> 3) In fact, when I first write this paper, I am struggling with the position of our method. Your comments make me think about it deeply. Now, we summarize our main as follows:
>
> * The assumption of "small perturbations are caused by the virtual movement filed" is a completely new idea in the literature of adversarial training or adversarial examples. By this assumption, we introduce data dependent constraints into the space of perturbations. And we cast the problem of finding perturbations into the problem of finding movement field.
>
> * We develop a general framework to design regularization terms for neural networks trained with lattice structured data, i.e. solving a min-max problem associated with the movement field.
>
> * To make above min-max problem easier, we introduce strong geometrical constraints into the movement field. Those constraints have two effects: first, it makes the adversarial movement field and the corresponding regularization term solved in closed-form which yields lower computational costs; second, it makes the obtained adversarial movement field has much more geometrical interpretability.
>
> The word “human perception” may be used inappropriately in the paper. However, we think the link between "interpretability" and “geometry” is obvious. For example, for VMT-shift, we can see which direction of movement of an image is most likely to fool the network. We can also see why VMT-scale is not suitable for synthetic dataset. (See section 4.1).
>
> 4) Sorry about the typos. We check them carefully in our updated paper.

---

> > ### Comment · AnonReviewer1 · 2018-01-12
> > **Comment about the revised rating**
> >
> > As, the authors have carefully responded to my comments I upgrade my score to 5. However, for eq. (23) it would be interesting to explain the motivation behind the choice of random sampling wi’s as a uniform random variables and compare to other ponderation choices.

---

> > > ### Author Response · Authors · 2018-01-16
> > > **About the chioce of w_i**
> > >
> > > Thanks for your higher score.
> > >
> > > We have tried two ways of setting wi's:
> > >  (1) Fixed values, i.e. wi = 1/3.
> > >  (2) Sampling wi' s from uniform distribution then normalizing them to a vector with unit length.
> > > And we find (2) is better than (1).
> > >
> > > Here are potential reasons:
> > > First, when we linearly combine different regularization operators, we get new operators in some sense. And when the operators are combined with different weights, we may say they are all different operators though they are highly related. Thus, compared with the fixed wi's, we can construct more regularization operators by randomly sampling wi's.
> > > Second, randomly sampling wi's may introduce noise into the regularization terms. As shown in previous works,  the generalization ability can be improved by introducing noise into the model in a proper way.
> > >
> > > Other choices, such as sampling wi's from different distributions may be better. But what we want to emphasize is that we believe randomly sampling wi's is better than fixing wi's.

---

### Official Review · AnonReviewer2 · 2017-11-29
**Train improvement by adversarial trail examples generated with differential motion fields**

**Rating:** 6
**Confidence:** 4

**Review:**

Summary:
The paper propose a method for generating adversarial examples in image recognition problems. The Adversarial scheme is inspired in the one proposed by Goodgellow  et al 2015 (AT) that introduces small perturbations to the data in the direction that increases the error. Such a perturbations are random (they have not structure) and lack of interpretation for a human user. The proposal is to limit the perturbations to just three kind of global motion fields: shift, centered rotation and scale (zoom in/out). Since the motions are small in scale, the authors use a first-order Taylor series approximation  (as in classical optical flow). This approximation allows to obtain close formulas for the perturbed examples; i.e. the correction factor of the Back-propagation computed derivatives w.r.t. original example. As result, the method is computational efficient respect to the AT and the perturbations are interpretable.
Experiments demonstrate that with the MNIST database is not obtained an improvement in the error reduction but a reduction of the computational time. However, with ta more general recognition problem conducted with the CIFAR-10 database, the use of the proposed method improves both the error and the computational time, when compared with AT and Virtual Adversarial Train.

Comments:

1. The paper presents a series os typos: FILED (title), obouve, freedm, nerual,; please check carfully.

2. The Derivation of eq. (13) should be explained, It could be said that (12) can be casted as a eigenvalue problem [for example: $ max_{\tilde v} \| \nabla_p L^T \tilde v \|^2 \;\; s.t. \| v\|=1  $] and (13) is the largest eigenvalue of $ \nabla_p L \nabla_p L^T $]

3. The improvement in the error results in the db CIFAR-10 is good enough to see merit in the proposal approach. Maybe other perturbations with closed formula could be considered and linear combinations of them

---

> ### Author Response · Authors · 2017-12-17
> **Authors' response to AnonReviewer2**
>
> Thanks for your recognition of our work.
> (1) Sorry about the typos. We have checked them carefully in our updated paper.
>
> (2) We provide the derivation of eq.(13) in Appendix C. It looks unnecessary to cast eq.(12) as an eigenvalue problem because there is just one unknown variable in eq.(12).
>
> (3) In fact, VMT-all is a linear combination of other three terms (See eq.(23)) and it achieves better performance compared with the individual term. We can except that the performance could be further improved if we design and combine more close-form terms and such combination is cheap in practice (See running time of VMT-all in Tab 3). However, we think 'shift', 'rotation' and 'scale' used in the paper are enough to show the merit of our method.

---

### Decision · Program_Chairs · 2018-01-29
**ICLR 2018 Conference Acceptance Decision**

**Decision:**

Invite to Workshop Track

**Comment:**

R1 thought the proposed method was novel and the idea interesting. However, he/she raised concerns with consistency in the experimental validation, the trade-off between accuracy and running time, and the positioning/motivation, specifically the claim about interpretability. The authors responded to these concerns, and R1 upgraded their score. R2 didn’t raise major concerns or strengths. R3 questioned the novelty of the work and the experimental validations. All reviewers raised concerns with the writing. Though I think the work is interesting, issues raised about experiments and writing make me hesitant to go against the overall recommendation of the reviewers, which is just below the bar. I think this is a paper that could make a good workshop contribution.